

# The effect of blow flies (Diptera: Calliphoridae) on the size and weight of mangos (*Mangifera indica* L.)

Shafqat Saeed[1,*], Muhammad Nadir Naqqash[2], Waqar Jaleel[3,*], Qamar Saeed[4] and Fozia Ghouri[3]

[1] Department of Entomology, Muhammad Nawaz Sharif University of Agriculture, Multan, Pakistan
[2] Department of Plant Production and Technologies, Faculty of Agricultural Sciences and Technology, Niğde University, Faculty of Agricultural Sciences and Technology, Nidge, Turkey
[3] College of Agriculture, South China Agriculture University, Guangzhou, China
[4] Department of Entomology, Faculty of Agricultural Sciences and Technology, Bahauddin Zakariya University, Multan, Pakistan
* These authors contributed equally to this work.

## ABSTRACT

**Background:** Pollination has a great effect on the yield of fruit trees. Blow flies are considered as an effective pollinator compared to hand pollination in fruit orchards. Therefore, this study was designed to evaluate the effect of different pollination methods in mango orchards.

**Methodology:** The impact of pollination on quantity and quality of mango yield by blow flies was estimated by using three treatments, i.e., open pollinated trees, trees were covered by a net in the presence of blow flies for pollination, and trees were covered with a net but without insects.

**Results:** The maximum number of flowers was recorded in irregular types of inflorescence, i.e., 434.80 flowers/inflorescence. Fruit setting (bud) was higher in open pollinated mango trees (i.e. 37.00/inflorescence) than enclosed pollination by blow flies (i.e. 22.34/inflorescence). The size of the mango fruit was the highest (5.06 mm) in open pollinated tree than those pollinated by blow flies (3.93 mm) and followed by without any pollinator (3.18 mm) at marble stage. We found that the maximum weight of mango fruit (201.19 g) was in open pollinated trees.

**Discussion:** The results demonstrated that blow flies can be used as effective mango pollinators along with other flies and bees. The blow flies have shown a positive impact on the quality and quantity of mango. This study will be helpful in future and also applicable at farm level to use blow flies as pollinators that are cheap and easy to rear.

Corresponding author
Shafqat Saeed,
bumblebeepak@gmail.com

# INTRODUCTION

The mango, *Mangifera indica* L., is very popular and economically important fruit. It is widely cultivated in the tropical and subtropical areas of the world (*Tjiptono et al., 1984*). Insect pollinators play a key role in maintaining global biodiversity by providing a key

ecosystem that is crucial for maintenance of domesticated and wild plant communities. Decline in pollinators' fauna and parallel shrinkage of dependent plants is well documented in scientific literature (*Potts et al., 2010*; *Natural Research Council, 2006*). According to an estimate, insect pollinators are responsible for 35% of global crop-based food production. About 87 crops, i.e. 70% of the 124 main crops used directly for human consumption in the world, have their pollination dependent upon pollinators (*Klein et al., 2007*). Recently, the global economic value of pollination from domesticated and wild animals has been estimated at €153 billion (*Gallai et al., 2009*). Moreover, destruction of natural habitats resulting in the decline of pollinators' fauna has laid the basis for discovery of the potential of more insect pollinators to increase the yield (*Hoehn et al., 2008*).

Unfortunately, research regarding pollinators usually focuses on hymenopterans, syrphids and butterflies. Therefore, all the ecology conservation schemes and other strategies are predominantly aimed to conserve these insects (*Potts et al., 2006*; *Potts et al., 2009*). Very little importance has been given to dipterans which consist of seventy-one families, including Mycetophilidae, Bibionidae, and Culicidae; Syrphidae, Bombyliidae, Conopidae, Stratiomyidae, and Nemestrinidae (lower Brachycera) and among the higher Brachycera (Cyclorrhapha); and many more (*Kastinger & Weber, 2001*; *Larson, Kevan & Inouye, 2001*; *Rotheray & Gilbert, 2011*). They contain regular visitors of more than 555 plant species (*Larson, Kevan & Inouye, 2001*). Non-syrphid Diptera are diverse, common and ubiquitous in both natural and managed habitats (*Skevington & Dang, 2002*; *Vanbergen et al., 2014*). Among non-syrphid dipterans, members of family Calliphoridae (Schizophora, Calyptratae, Oestroidea) commonly known as blow flies, bluebottles, cluster flies or greenbottles are very important pollinators. They are distributed worldwide, with over 1,000 species and about 150 genera described (*Wood, 1987*; *Brown et al., 2010*). Blow flies is thought to be the most disliked fly among all the flies of Dipteran, and it is a carrier for the most diseases and causes myiasis (*Zumpt, 1965*; *Greenberg, 1973*). It was recognized for nearly 1,500 years ago that flies are transmitters of diseases (*Greenberg, 1973*). Early research was done only on the negative aspects of flies, but now most of the studies have shown that blow flies species have many beneficial aspects such as surgeons, pollinators, agents of decay, forensic indicators, and recreational uses (*Jarlan, de Oliveira & Gingras, 1997*; *Losey & Vaughan, 2006*; *Klein et al., 2007*; *Heath, 2015*). *Kugler (1950)* and *Kugler (1951)* conducted series of experiments and demonstrated that green bottle flies, blow flies and flesh flies (*Sarcophaga* sp.) favor yellow and white colored models over brown and purple ones in the presence of a sweet scent, but the opposite in the presence of a carrion scent while unscented models were ignored. The survey by *Kumari et al. (2014)* was done in a mango orchard during different times and reported that blow flies performing as pollinators in mango orchards gave better yield as compared to unpollinated trees.

The biology of mango pollinators have been studied in India and Israel, and their results demonstrated that insects of the Diptera and Hymenoptera play major roles in the pollination of this important fruit. Important examples of mango pollinators' are e.g. *Apis dorsata* F., *Apis florea* F., *Episyrphis balteatus* De Geer., *Ischiodon scutellaris* F. *and*

*lucilia spp* (*Singh, 1988*; *Bhatia et al., 1995*; *Singh, 1997*; *Dag & Gazit, 2000*). Use of these pollinators can significantly enhance the mango yield (*Anderson et al., 1982*; *Dag & Gazit, 2000*; *Rafique et al., 2016*). Considering the importance of beneficial aspects of blowflies in Pakistan, the role of pollinators, especially dipterans (blow flies), were never studied in *Mangifera indica*. The blowflies are the easier source of pollination as compared to other pollinators such as honey bees, syrphid flies and *Xylocopa spp* that are difficult to rear (*Faulkner, 1977*). Moreover, all the stages of fruit development like formation of buds, pea size, marble size fruit (weight and size) and stone stage are very important with respect to development and economic yield of mangos so impact of only blow flies and all pollinators was investigated on these stages (*Verghese, 1999*). Considering the importance of beneficial aspects of blow flies as role of pollinators especially which has been never studied in *Mangifera indica*. Therefore, this research was conducted to evaluate the effects of blow flies on the mango pollination and fruit yield and quality. Blow flies are the cheapest source of pollination as compared to other pollinators, such as honey bees, syrphid flies, *xylocopa spp* that are expensive to rear.

## MATERIAL AND METHODS

### Plant material

The impact of pollination by blow fly on mango yield was studied in the orchard of the Faculty of Agricultural Science and Technology (FAS&T), Bahauddin Zakariya University, Multan. A total of three trees and 10 branches from each tree were selected for recording the data. The following treatments were used: (1) open pollinated trees; (2) trees were covered by net and blow flies were used for pollination; (3) trees were covered by nets and no insect was kept inside the net for pollination. Three replications were used for each treatment.

### Rearing of blow flies for mango pollination

Adults of blow fly (*Calliphora* spp.) were collected from the different poultry farms of Multan, Pakistan. Mass culture of blow flies was reared in the Bio-Ecology Laboratory of Faculty of Agricultural Science and Technology, Bahauddin Zakariya University, Multan Pakistan. Adults were released into the plastic cage (18 cm in diameter and 24 cm in height) with diet (10% honey solution), and chicken livers were also placed in the plastic trays for egg laying. The six plastic cages were used for rearing blow flies. Then hatched larvae were separated into the plastic pots (4 cm in diameter and 8 cm in height) that were half filled with sterilized sand and 50 g chicken liver. In each plastic pot, 20 larvae were released and maximum adults of blow flies were reared in the laboratory for field application.

### Installation of cages

Experiment was installed according to randomized complete block design (RCBD). Mango trees with a height of 2.1 m and width of 2.4 m at the emergence of inflorescence were selected for the installation of cages. The cages, made by muslin cloths, were used for the covering of mango trees (3.35 × 3.35 × 3.35 m). A total of 100 adult

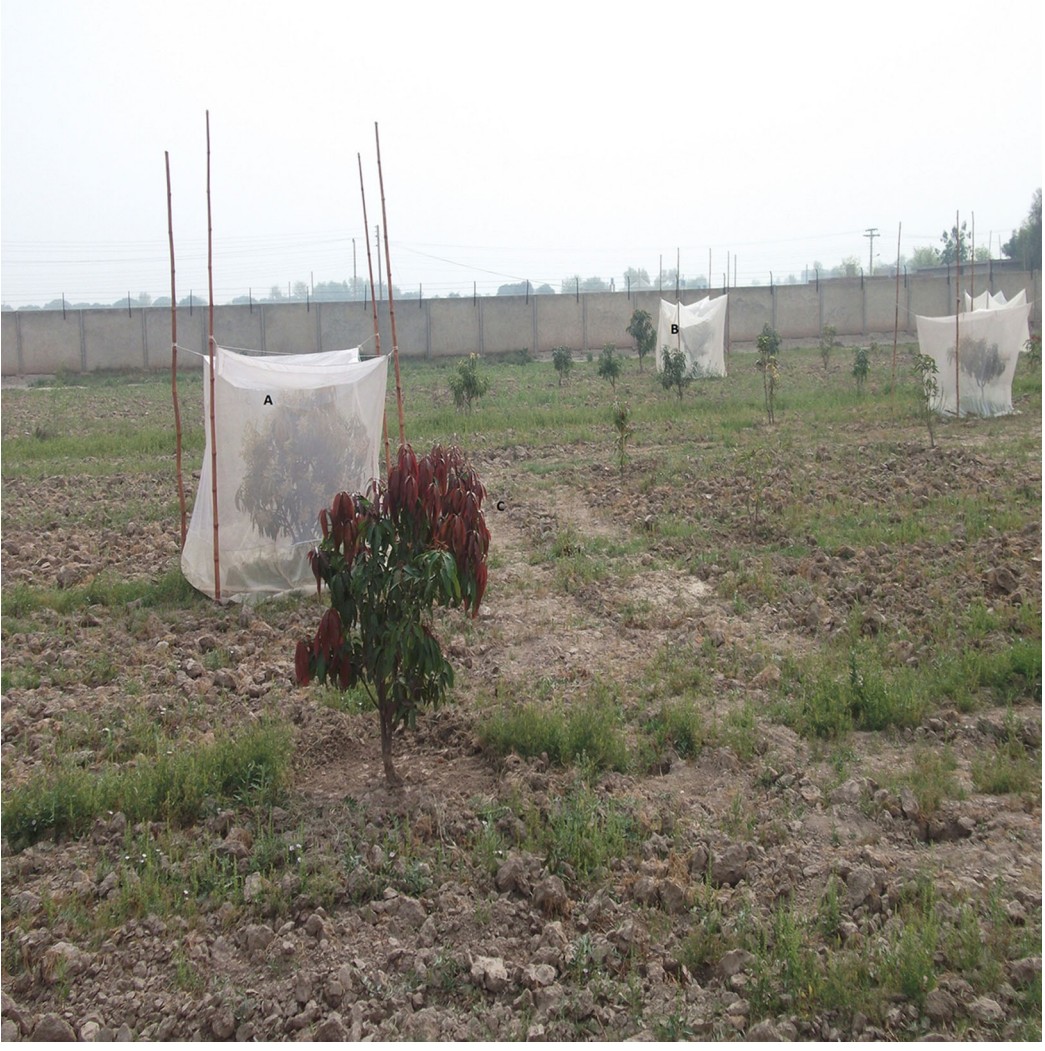

**Figure 1 Open pollinated and covered trees in a mango orchard.** (A) Mango trees covered with a muslin cloth and without pollinator; (B) Mango trees covered with a muslin cloth in the presence of blow flies; and (C) open trees for different pollinators.

blow flies were released for pollination efficacy in the covered mango trees (Fig. 1) and control was free of all kinds of insect pollinators. Ten branches were tagged in each tree for data recording.

## Data recording

Total number of flowers and their types of inflorescences were counted in each treatment. Data as number of flowers on each type of inflorescence, size (mm) and weight (g) of fruits at marble stage (30 days after the fruit set and have no stone (stage of seed) inside the fruit) of mangos was recorded by tagging ten twigs in each repeating unit.

## Statistical analysis

The data regarding type of the inflorescence (i.e., conical, pyramid and irregular) in three different treatments (open tree, blow fly cage, without pollinators/flies/close cage),

**Table 1  Total number of inflorescences and their types in different treatments.**

| Treatments | Trees | Number and types of mango inflorescence | | | |
|---|---|---|---|---|---|
| | | Conical | Pyramid | Irregular | Total |
| Closed | Tree 1 | 13.00 | 18.00 | 22.00 | 53.00 |
| | Tree 2 | 15.00 | 34.00 | 20.00 | 69.00 |
| | Tree 3 | 34.00 | 19.00 | 33.00 | 86.00 |
| Blow flies | Tree 1 | 19.00 | 21.00 | 21.00 | 61.00 |
| | Tree 2 | 11.00 | 13.00 | 12.00 | 36.00 |
| | Tree 3 | 20.00 | 34.00 | 20.00 | 74.00 |
| Open | Tree 1 | 45.00 | 50.00 | 30.00 | 125.00 |
| | Tree 2 | 27.00 | 23.00 | 26.00 | 76.00 |
| | Tree 3 | 23.00 | 43.00 | 20.00 | 86.00 |

numbers of flowers, buds, fruits, size and weight of fruits were subjected to statistical analysis using analysis of variance (ANOVA). Means were compared by use of least significance difference test at $P = 0.05$. Data were analyzed using SAS program (SAS Institute, Cary, NC, USA).

## RESULTS

Number of inflorescences was initially counted before pollination/treatment among each type of inflorescence (conical, pyramid and irregular) (Table 1). The number of opened flowers ($327.97 \pm 25.92$) was significantly higher in open pollinated trees as compared to blow fly and unpollinated cages ($F = 80.04$; $DF = 6$; $P < 0.01$). While on irregular type of inflorescence, the number of opened flowers in the open pollinated mango trees was more i.e. $434.80 \pm 52.30$/inflorescence than in unpollinated and blow fly pollinated trees ($F = 14.06$; $DF = 6$; $P < 0.01$). Similar pattern was recorded regarding numbers of opened flowers in pyramid type of inflorescence ($F = 54.06$; $DF = 6$; $P < 0.01$) (Fig. 2).

The highest number of buds/inflorescence were found in open mango pollinated trees ($2.67 \pm 0.51$/inflorescence) than the blow fly cage and the cage without pollinators, in case of conical inflorescence ($F = 13.3$; $DF = 6$; $P < 0.01$). Data regarding pyramid type of inflorescence depicted that maximum numbers of buds ($1.96 \pm 0.45$/inflorescence) were observed in open pollinated trees than blow fly and without pollinators cage trees ($F = 8.09$; $DF = 6$; $P < 0.01$). Similar pattern of results was found in case of irregular inflorescence where higher number of buds i.e. $1.93 \pm 0.47$/inflorescence were found as compared to rest of two treatments ($F = 15.02$; $DF = 6$; $P < 0.01$) (Fig. 3). A significantly higher number of buds on conical inflorescence i.e. $3.53 \pm 0.05$/inflorescence were found in open trees than in the blow fly cage and unpollinated cages ($F = 10.08$; $DF = 6$; $P < 0.001$). Similarly, the number of buds on pyramid inflorescence was significantly higher, i.e. $3.60 \pm 0.34$/inflorescence in open-pollinated trees than in blow fly pollinated and unpollinated cages ($F = 17.07$; $DF = 6$; $P < 0.01$). In cases of irregular inflorescence, a higher number of buds/inflorescences were found in blow fly pollinated trees ($4.16 \pm 0.11$/inflorescence) than open and unpollinated trees ($F = 20.06$; $DF = 6$; $P < 0.01$) (Fig. 4).

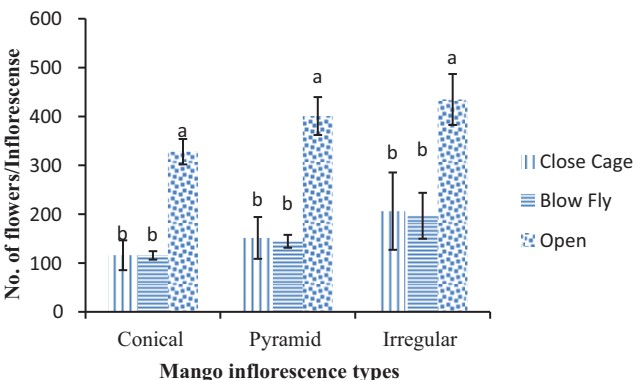

**Figure 2 Effect of different pollination methods on the different types of mango flowers.** Mean values sharing similar letters did not differ significantly within the treatments (P ≤ 0.001). Bars indicate the standard deviation (SD) of the observation.

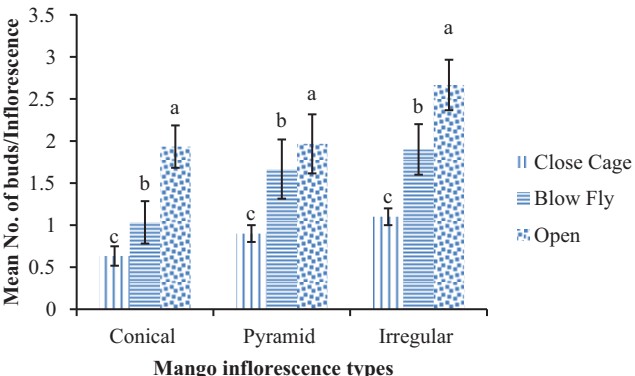

**Figure 3 Effect of different pollination methods on the bud formation/inflorescence after 15 days of treatment.** Mean values sharing similar letters did not differ significantly within the treatments (P ≤ 0.001). Bars indicate the SD of the observation.

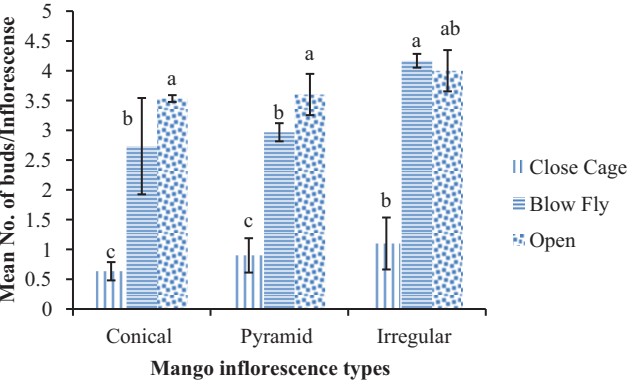

**Figure 4 Effect of different pollination methods on the bud formation/inflorescence after 10 days after the treatments.** Mean values sharing similar letters did not differ significantly within the treatments (P ≤ 0.001). Bars indicate the SD of the observation.

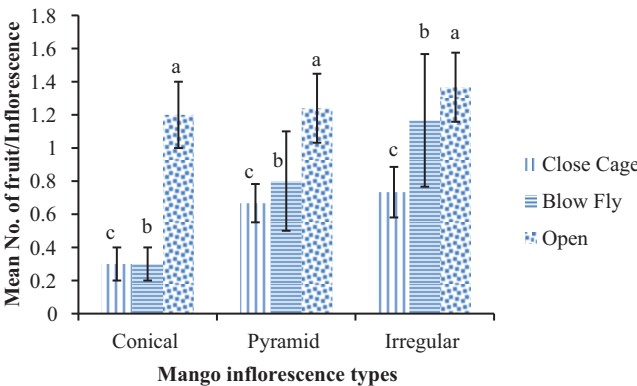

**Figure 5 Effect of different pollination methods on the number of fruits at marble stage.** Mean values sharing similar letters did not differ significantly within the treatments (P ≤ 0.001). Bars indicate the SD of the observation.

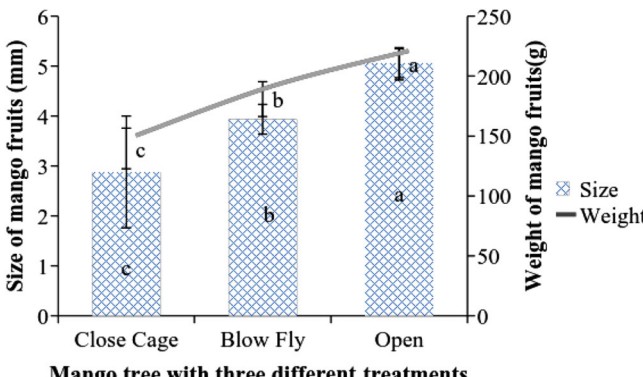

**Figure 6 Effect of different pollination methods on the fruit size and weight at marble stage.** Mean values sharing similar letters did not differ significantly within the treatments (P ≤ 0.001). Bars indicate the SD of the observation.

A significant difference was recorded in mango fruit formation at marble stage among three tested treatments in each type of inflorescence (conical, pyramid and irregular inflorescence). The number of fruits was significantly higher with conical inflorescence (1.20 ± 0.20/inflorescence) than in blow fly and close cage/without pollinators trees (F = 40.5; DF = 6; P < 0.05). Number of fruits (1.24 ± 0.21/inflorescence) at marble stage on pyramid inflorescence was more in open pollinated trees than other two treatments (F = 5.38; DF = 6; P < 0.01). A higher number of fruits/inflorescences was found in open pollinated mango trees, at marble stage, on irregular inflorescence (1.36 ± 0.20/inflorescence) than in blow fly pollinated and unpollinated cages of trees (F = 5.90; DF = 6; P < 0.01) (Fig. 5).

The average size and weight of mango fruits at marble stage varied significantly among the treatments. The average size of mango fruits, i.e. 5.06 ± 0.29 mm, was statistically higher in open pollinated trees than in blow fly pollinated cages and closed cages trees (F = 7.47; DF = 6; P < 0.01). A similar pattern was also observed in the case of weight of mango fruits where average weight of each mango i.e. 210.20 ± 13.92 g, was

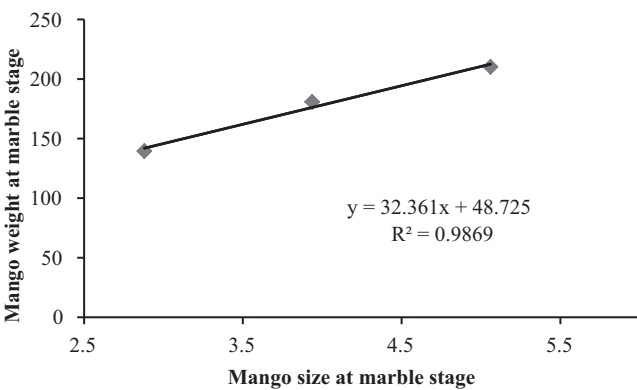

$y = 32.361x + 48.725$
$R^2 = 0.9869$

**Figure 7 Correlation between mango fruit size and weight at marble stage.**

significantly higher as compared to that in blow fly pollinated cages and closed cages trees (F = 16.07; DF = 6; P < 0.01) (Fig. 6). Data showed the positive correlation between of size and weight of mango fruit at the marble stage (Fig. 7).

## DISCUSSION

A huge number of economically nutritive plants depend on different types of pollinators for pollination (*Eilers et al., 2011*). Modern farming techniques can produce a higher yield of crops (*Aizen et al., 2008*; *Aizen et al., 2009*), but due to a significant decline in insect pollinators primarily due to the isolation from natural habitats, the potential of modern farming techniques can't be fully utilized (*Klein et al., 2007*; *Garibaldi et al., 2011*). In most of the habitats, pollinating flies guarantee or enhance seed and fruit production of many plants such as medicinal, food and ornamental plants. Due to the large gaps in the knowledge about the dipterans, there is a need to address the role of diptera in pollination network. Dipteran flies have potential to survive in variable ranges of temperature or environment our results support that blow fly have potential of pollination (*Ssymank et al., 2008*; *Munawar et al., 2011*; *Abrol, 2012*).

A mango panicle contains around 200–4,000 flowers, and a mature tree may have approximately 600–1,000 panicles (*Manning, 1995*). About 46 kinds of pollinators belonging to three orders i.e. coleoptera, diptera and hymenoptera are capable of pollinating mango flowers (*Singh, 1988*; *Bhatia et al., 1995*; *Singh, 1997*; *Dag & Gazit, 2000*). Mango inflorescence are three typical types: conical, irregular and various insects visited for pollination (*Heard, 1999*). These pollinators are very crucial for a successful fruit set in mangos (*Free & Williams, 1976*; *Anderson et al., 1982*; *Richards, 2001*; *Carvalheiro et al., 2010*). They are not only sensitive to change in their natural habitat and/or niche, but are also sensitive to pesticides (*De Siqueira et al., 2008*).

Pollinators are one of the main components of the ecosystem (*Garibaldi et al., 2013*). There are two types of pollinators: domesticated and wild pollinators; both are very important for the pollination of plants. Our result showed that open trees produced a maximum yield, followed by covered trees with blow flies and without insects. These results are consistent with the previous study, which revealed that insects increase the yield of fruits by amplifying pollination (*Mingjian, Zi & Jianguo, 2003*). Previous studies also

demonstrated that the diversity of pollinators has a greater impact on the yield of fruit trees, and that environmental hazards have reduced the different types of pollinators (*Jones & Emusweller, 1934*; *Fajardo et al., 2009*). In an open pollinated condition, mango size and weight were highly significant because of a variety of pollinators, e.g. *Apis dorsata* F., *Apis florea* F., *Episyrphis balteatus* De Geer., *Ischiodon scutellaris* F. and *lucilia spp* at the farm (*Bashir, Saeed & Sajjad, 2013*). The overall results showed that open pollinated trees yielded a maximum amount of fruit of good quality. However, blow flies were also shown to have a great impact on mango pollination because higher quantity and better quality of fruits were recorded than in close cage trees.

## CONCLUSION

The results revealed that fruit setting was better in open trees than blow flies and without pollinated trees, respectively. Mango weight and size of was significantly better in open trees than blow flies and without pollinators. However, we detected fruits with maximum weight and size in the open pollinated mango trees where a greater number of pollinators visit the trees for pollination and resulted in the better quality and quantity of mango fruit. We concluded that blow flies have the potential for pollination in *M. indica*. Therefore, this research will be helpful in the future and will be a applicable at the farm level where keeping honey in the orchard is difficult for pollination because of the environment and the high cost. We speculated that blow flies are the best, cheapest source of pollination as a replacement for honey bees and other pollinators which are expensive to purchase and to maintain in the orchards for pollination. Blow flies would be easily maintained in orchards by providing a diet of organic matter, dead birds etc, in one side of the mango orchard. This study also showed that irregular types of inflorescence lead to the maximum number of flowers, buds and fruits, so breeders could focus on developing the varieties of *M. indica* by having a greater number of irregular types of inflorescence.

### Funding
The authors received no funding for this work.

### Competing Interests
The authors declare that they have no competing interests.

### Author Contributions
- Shafqat Saeed conceived and designed the experiments, analyzed the data, contributed reagents/materials/analysis tools, wrote the paper, prepared figures and/or tables, reviewed drafts of the paper, provided support for arranging field for research.
- Muhammad Nadir Naqqash conceived and designed the experiments, performed the experiments, analyzed the data, wrote the paper, prepared figures and/or tables, reviewed drafts of the paper.

- Waqar Jaleel conceived and designed the experiments, performed the experiments, analyzed the data, contributed reagents/materials/analysis tools, wrote the paper, prepared figures and/or tables, reviewed drafts of the paper.
- Qamar Saeed conceived and designed the experiments, analyzed the data, contributed reagents/materials/analysis tools, wrote the paper, prepared figures and/or tables, reviewed drafts of the paper.
- Fozia Ghouri conceived and designed the experiments, performed the experiments, analyzed the data, wrote the paper, prepared figures and/or tables, reviewed drafts of the paper.

### Data Deposition

The raw data has been supplied as a Supplemental Dataset File.

### Supplemental Information

Supplemental information for this article can be found online at http://dx.doi.org/10.7717/peerj.2076#supplemental-information.

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
