# Peer review of "The effect of blow flies (Diptera: Calliphoridae) on the size and weight of mangos (Mangifera indica L.)"

_PeerJ, doi:10.7717/peerj.2076_

## Round 0.1 · original submission · Major Revisions

· Academic Editor

Major Revisions

Please look at all comments from the reviewers, but most especially the questions and suggestions from reviewer one regarding statistical analysis. It is crucial that you address these points. The reviewer's concern regarding only one trial relates to standard practices in agronomic publications which typically require more than one season's data for studies involving yield (to allow for potential environmental variability). In this paper, I don't think having more than one trial (=experiment) should be a barrier to publication. I look forward to seeing your revised manuscript.

Reviewer 1 ·

Basic reporting

The study is designed to show that blow flies can effectively pollinate and increase fruit yield of mango. The study was a cage trial with 3 treatments arranged in 3 replications. The study had open an pollinated plant treatment and cages with and without blow flies added. There was only one trial conducted which is a concern. The measurements seem appropriate, flowers, buds, and fruit size and weight. The main issue with the paper is the lack of description of the statistical procedures. It is not clear what the experimental design was; randomized complete block or completely randomized design? In addition, there is a complete the lack of indication of significance of treatments in the various figures with no means separation letters or F tests provided in the text of figures. The bar charts with SD marked do suggest there were treatment differences especially in figures between open plants and plants in cages with no insects, but differences between plants in cages with and without flies are less apparent. The only figure where a clear difference between cages with and without flies is figure 3a, no. of buds/inflorescence. Furthermore, the critical figure of fruit size and weight, it is not clear that numeric increases in fruit size and weight with flies over cages without flies are statistically significant. The weight values need to be presented in a separate chart with SD or SEM values.

Specific comments are as follows:
Title and text: It is standard convention to list fly as a separate word in the name of a true fly like blow fly or green bottle fly.
Abstract Line 13: delete ‘other’ bee. Blow fly is not a bee.
L. 62: What does a ‘consumer surplus loss’ mean?
L. 66 – 82: More information on previous work about pollination of mangoes including the line in the discussion of different flowers and pollinators (L. 173-174). Previous studies on pollination by dipterans and specifically blow flies also should be added to the introduction.
L. 90. Also nowhere in the text do the author’s list the species of blow fly used in the study.
L. 91. Add location
L. 113. Is a stone the seed?
L. 116-119: What was the experimental design? What SAS procedure was used for the analysis of variance? Was a mixed model used? If yes, what were the fixed and random effects?
L. 125-126: This sentence is awkward and unclear as written.
L. 127: Are these flower buds or fruit buds?
L. 143-150: Again cannot tell what is actually statistically significant because the means separate letters and values are not presented. The only really obvious result is that cages without flies clearly reduce fruit size and weight as compared with an open situation with access to local pollinators.
L. 144-145: Need to add some measure of variation to the mean numbers. I would suggest listing them in a figure or table, not in the text.
L. 146-147. Add author to species names at first use.
L. 159-160. Not sure why this statement is relevant to the study.
L. 164-165: This statement may or may not be true depending on the statistical results.
L. 169: Have all of these orders of insects been shown to pollinate mangoes?
L. 173-174. Interesting. This should be discussion in the introduction. What are the different kinds of flowers and how are various insects important to pollination? Next sentence, ‘These pollinators’ what are the specific pollinators.
L. 178-180. These sentences are not clear and should be re-written.
L. 187-188. Delete this sentence.
L. 189. Is a separate conclusion or summary necessary?

Experimental design

The study was a cage trial with 3 treatments arranged in 3 replications. The study had open an pollinated plant treatment and cages with and without blow flies added. There was only one trial conducted which is a concern. The measurements seem appropriate, flowers, buds, and fruit size and weight. The main issue with the paper is the lack of description of the statistical procedures. It is not clear what the experimental design was; randomized complete block or completely randomized design? In addition, there is a complete the lack of indication of significance of treatments in the various figures with no means separation letters or F tests provided in the text of figures. The bar charts with SD marked do suggest there were treatment differences especially in figures between open plants and plants in cages with no insects, but differences between plants in cages with and without flies are less apparent. The only figure where a clear difference between cages with and without flies is figure 3a, no. of buds/inflorescence. Furthermore, the critical figure of fruit size and weight, it is not clear that numeric increases in fruit size and weight with flies over cages without flies are statistically significant. The weight values need to be presented in a separate chart with SD or SEM values.

L. 116-119: What was the experimental design? What SAS procedure was used for the analysis of variance? Was a mixed model used? If yes, what were the fixed and random effects?

Validity of the findings

Only 1 field trial is a concern.
Lack of means separation in the figures and F or t test values and details about the statistical procedures makes assessment of the results impossible.

Additional comments

Assuming the statistical result do show that there is increased fruit size and weight in the presence of blow flies, how will this be used in the field? Presumably mango farmers are not going to cage plants. In an open setting will blow flies remain in the mango field long enough to pollinate the trees or will they disperse as soon as released?

Reviewer 2 ·

Basic reporting

Thanks for the submission of the manuscript. It is a concise and easy to read paper.

The biggest areas of improvement in the introduction of the background is the need 1) for a better link the need of pollination due to pollinator declines (or suspected declines). 2) You need to describe some of the pollinators of mangos - (what are the main groups)? These should be listed and described. 3) Known effect of these pollinator groups on yield of - much more is needed here . 4) Please strengthen the why blowflies can be used for pollination of mangos, you mention cheap to rear. What else? Also, do they spread any disease/bacteria/or virus to the mango fruit? 5) Talk a little about fruit stages in the background and the why you need to investigate and record impacts on the different stages of fruit development.

Some aspects of the discussion belong in the Introduction and not in the discussion.

Experimental design

The experimental design is good generally. Did you also look at what insects visited the mangos in the open pollination? Please report this. This is critical How do we know any other pollinators visited the mangos? Maybe different species of blowflies were the only open pollinators on mangos?

What species of blowflies were used in the closed pollination? This is very important.

Validity of the findings

You did not report any statistics with Means and SD, F, or p values for your experiment. This must be done in the results section.

It is hard to validate results without the aforementioned comments in Experimental design and the statistics being addressed.

Additional comments

This is an interesting idea that needs some holes-filled in the writing, results, and discussion (see comments within) before it is ready for publication.

Some minor grammar things throughout that can be addressed if re-reviewed allowed.

---

## Round 0.2 · accepted · Accept

· Academic Editor

Accept

Thanks for your revision, and I especially appreciate your detailed response to the reviewers comment, as it made my job very easy. I think the revised manuscript looks very good, and I'm happy to accept it for publication.